# Genome-Wide Identification of the U-Box E3 Ubiquitin Ligase Gene Family in Cabbage (*Brassica oleracea* var. *capitata*) and Its Expression Analysis in Response to Cold Stress and Pathogen Infection

**DOI:** 10.3390/plants12071437

**Published:** 2023-03-24

**Authors:** Peiwen Wang, Lin Zhu, Ziheng Li, Mozhen Cheng, Xiuling Chen, Aoxue Wang, Chao Wang, Xiaoxuan Zhang

**Affiliations:** 1College of Horticulture and Landscape Architecture, Northeast Agricultural University, Harbin 150030, China; wangpeiwen@neau.edu.cn (P.W.); 18645096256@163.com (L.Z.); lzh18236452381@163.com (Z.L.); mzcheng@neau.edu.cn (M.C.); chenx@neau.edu.cn (X.C.); axwang@neau.edu.cn (A.W.); wangchao504@126.com (C.W.); 2Key Laboratory of Biology and Genetic Improvement of Horticultural Crops (Northeast Region), Ministry of Agriculture and Rural Affairs, Northeast Agricultural University, Harbin 150030, China

**Keywords:** cabbage, plant U-box E3 ubiquitin ligase (PUB), gene family, phylogenetic analysis, gene expression

## Abstract

Plant U-box E3 ubiquitin ligases (PUBs) play an important role in growth, development, and stress responses in many species. However, the characteristics of U-box E3 ubiquitin ligase genes in cabbage (*Brassica oleracea* var. *capitata*) are still unclear. Here, we carry out the genome-wide analysis of U-box E3 ubiquitin ligase genes in cabbage and identify 65 *Brassica oleracea* var. *capitata* U-box E3 ubiquitin ligase (BoPUB) genes in the cabbage genome. Phylogenetic analysis indicates that all 65 *BoPUB* genes are grouped into six subfamilies, whose members are relatively conserved in the protein domain and exon-intron structure. Chromosomal localization and synteny analyses show that segmental and tandem duplication events contribute to the expansion of the U-box E3 ubiquitin ligase gene family in cabbage. Protein interaction prediction presents that heterodimerization may occur in BoPUB proteins. In silico promoter analysis and spatio-temporal expression profiling of *BoPUB* genes reveal their involvement in light response, phytohormone response, and growth and development. Furthermore, we find that *BoPUB* genes participate in the biosynthesis of cuticular wax and in response to cold stress and pathogenic attack. Our findings provide a deep insight into the U-box E3 ubiquitin ligase gene family in cabbage and lay a foundation for the further functional analysis of *BoPUB* genes in different biological processes.

## 1. Introduction

Ubiquitin-mediated protein degradation is an essential regulatory mechanism in the growth and immunity of eukaryotes, such as cell cycle [1], signal transduction [2], and immune response [3]. In this process, substrate proteins are tethered with a polyubiquitin chain by the sequential catalysis of three enzymes, ubiquitin-activating enzyme (E1), ubiquitin-conjugating enzyme (E2), and ubiquitin ligase (E3) [4], and the resulting ubiquitin-conjugated proteins are then recognized and degraded by the 26S proteasome [4]. The E3 ubiquitin ligase responsible for the recognition specificity of substrates is considered to be a crucial component in the ubiquitin-proteasome system [4,5]. E3 ubiquitin ligases have been extensively described in fungi [6], plants [7], and humans [8]. In plants, E3 ubiquitin ligases are divided into two types based on the number of their subunits, including monomeric E3 ubiquitin ligase and multi-subunit E3 ubiquitin ligase complex. According to the variety of subunits, monomeric E3 ubiquitin ligases are grouped into three subtypes: homology to E6-AP C-terminus (HECT), really interesting new gene (RING), and U-box E3 ubiquitin ligases [7]; and multimeric E3 ubiquitin ligases are grouped into two subtypes: cullin-RING ligase (CRL) and anaphase promoting complex/cyclosome (APC/C) E3 ubiquitin ligase complexes [7]. A variety of plant E3 ubiquitin ligases contribute to their functional diversity in a wide range of biological processes, including light response [9], phytohormone regulation [10], stress response [11], and senescence [12].

Plant U-box E3 ubiquitin ligases (PUBs) are a class of U-box domain-contained proteins. The U-box domain consisting approximately 70 amino acids structurally resembles the RING finger domain except for the lack of conserved cysteine and histidine residues chelating two zinc ions [13]. The U-box domain highly conserved in eukaryotes is used for docking E2-ubiquitin conjugates and is involved in dimer formation, which is crucial for its protein activity [13]. In contrast to U-box E3 ubiquitin ligases in yeast and human, the combination of U-box domain and additional motifs related to protein interaction is commonly found in PUBs [14,15]. These plant-specific U-box E3 ubiquitin ligases are likely to be involved in response to environmental stimuli unique to sessile organisms. A large number of genetic analyses in *Arabidopsis* and rice have demonstrated that PUBs are implicated in development and stress responses [7,13]. For instance, loss of *AtPUB4* function promotes cell proliferation in shoot and root apical meristems [16], and the transcript level of *OsUPS*, a PUB gene from rice, is up-regulated under phosphate starvation conditions [17]. In recent decades, ever-increasing evidence has revealed that PUBs can also serve as key regulators in horticultural plants, including *Brassica napus* [18], *Solanum lycopersicum* [19], and *Malus domestica* [20]. So far, the PUB gene family members have been well identified in many horticultural plants, including *Brassica rapa* [21], *Vitis vinifera* [22], *Solanum lycopersicum* [23], *Pyrus bretschneideri* [24], and *Brassica oleracea* [25]. Previous studies have indicated that 99 PUB gene family members are identified in *Brassica oleracea* and divided into seven clades, and transcriptome analysis has shown that these members are likely to participate in pollination [25]. However, the expression patterns of PUB genes in *Brassica oleracea* under stress conditions are rarely reported.

As an essential member in the *Brassica* family of horticultural plants, *Brassica oleracea* includes many important vegetable crops, including cauliflower, broccoli, cabbage, brussels sprout, kohlrabi, and kale, and is widely grown throughout the world [26]. Moreover, *Brassica oleracea* is also considered to be a model for the study of the genomic evolution of *Brassica* species due to its typical diploid genome structure [26]. The molecular mechanisms underlying many significant biological processes, including pigment accumulation [27], self-incompatibility [28], and flowering [29], have been dissected in *Brassica oleracea*, while a lot of classic biotechnologies, including virus-induced gene silencing (VIGS) [30] and CRISPR/Cas9-based gene editing [31], have also been applied in *Brassica oleracea*. Thus, the discovery of functional genes in *Brassica oleracea* is required for the genetic improvement of desired traits by the most advanced molecular breeding strategies.

With the development of bioinformatics in recent decades, a high-quality genome sequence of cabbage (*Brassica oleracea* var. *capitata*) has been presented [26], which dramatically accelerates the characterization of various gene families in cabbage, such as polygalacturonase [32] and WRKY transcription factor gene families [33]. However, little is known about the U-box E3 ubiquitin ligase gene family in cabbage and its functions in response to biotic and abiotic stresses. In the present study, a total of 65 *Brassica oleracea* var. *capitata* U-box E3 ubiquitin ligase (BoPUB) genes are confidently found in the cabbage genome. The phylogenetic relationships, conserved domains, exon-intron structures, chromosomal localization, and duplication events of these *BoPUB* genes are characterized. Interaction relationships between BoPUB proteins, cis-regulatory elements in the promoters of *BoPUB* genes, and their spatio-temporal expression are investigated. Moreover, we also analyze the expression levels of *BoPUB* genes involved in cuticular wax biosynthesis and in response to cold stress and pathogen infection. Our findings provide important guidance for the further functional analysis of U-box E3 ubiquitin ligase genes in cabbage.

## 2. Results

### 2.1. Genome-Wide Identification and Phylogenetic Relationships of BoPUB Genes

To identify U-box E3 ubiquitin ligase genes in cabbage, the protein sequence of *Arabidopsis thaliana* U-box E3 ubiquitin ligase carboxy terminus of Hsc70-interacting protein (AtCHIP) was used as queries to perform a retrieval against the cabbage protein database. Meanwhile, the hidden Markov model (HMM) profiles of U-box domain (PF04564.15) were also used to identify U-box E3 ubiquitin ligase genes in cabbage using the HMMER 3.0 software. In total, 65 U-box E3 ubiquitin ligase genes were confidently found in the cabbage genome and named after *BoPUB1-65* according to their gene IDs (Appendix A). To analyze the phylogenetic relationships of *BoPUB* genes, a phylogenetic tree based on the sequence alignment of BoPUB proteins was constructed by the neighbor-joining (NJ) method with 1000 bootstrap replicates (Figure 1 and Appendix A). All 65 *BoPUB* genes were grouped into six subfamilies (Group A-F) containing 16, 9, 9, 4, 20, and 7 members, respectively (Figure 1). Compared with the previous study [25], we identified fewer U-box E3 ubiquitin ligase genes in cabbage, probably because we used a rigorous gene screening method that completely excluded putative *BoPUB* genes containing RING finger domains. The characteristics of these *BoPUB* genes were then described, including genome position, the length of coding sequence (CDS), the number of amino acids (AA), protein molecular weight (MW), protein isoelectric point (pI), and subcellular localization (Appendix A). The coding sequence length of each *BoPUB* gene ranged from 921 bp (*BoPUB55*) to 3198 bp (*BoPUB32*), and the number of amino acids in each BoPUB protein sequence ranged from 307 AA (BoPUB55) to 1066 AA (BoPUB32) (Appendix A). The molecular weight of each BoPUB protein varied from 34.45 kDa (BoPUB55) to 118.1 kDa (BoPUB32), and the isoelectric point of each BoPUB protein varied from pH 4.68 (BoPUB18) to pH 9.85 (BoPUB43) (Appendix A). The prediction of subcellular localization showed that most of the BoPUB proteins (63%) belonged to nuclear and cytoplasmic proteins, and the other members were related to the plasma membrane, endoplasmic reticulum (ER), and chloroplast (Appendix A).

### 2.2. Conserved Domains and Exon-Intron Structures of BoPUB Genes

To analyze the composition of conserved domains in BoPUB proteins, the amino acid sequences of BoPUB proteins were submitted to the Pfam database (http://pfam.xfam.org/ accessed on 20 March 2022). Thirteen conserved domains were identified in the BoPUB protein family (Figure 2a,b and Appendix A), including U-box domain, armadillo (Arm) repeat, Arm_2 repeat, ADP ribosylation factor (Arf) domain, Prp19 domain, WD40 repeat, ubiquitin elongating factor core (Ufd2P_core) domain, tetratricopeptide repeat (TPR)_8 domain, protein kinase (Pkinase) domain, protein tyrosine and serine/threonine kinase (PK_Tyr_Ser-Thr) domain, universal stress protein (Usp) domain, CTLH/CRA C-terminal to LisH motif (CTLH) domain, and GDA1/CD39 nucleoside phosphatase (GDA1_CD39) domain. The U-box domain responsible for protein activity was detected in all BoPUB members, and the other domains were distributed in different BoPUB protein subfamilies according to evolutionary relationships (Figure 2a,b). All the members in Group A only contained a U-box domain: sixteen members (73%) in Group B-D had Arm/Arm_2 repeats; seven members (35%) in Group E carried Pkinase/PK_Tyr_Ser-Thr/Ufd2P_core domains; and two members (29%) in Group F included Arm repeats (Figure 2a,b). Similar to other cruciferous plants, *BoPUB* genes with ARM repeats remain the largest component of the PUB gene family members in cabbage [21,25]. In addition to some classic domains (WD40, TRP, and Pkinase), we also identify several special domains, including Arf, CTLH, and Usp, that are not commonly detected in previous studies on the PUB gene family in cruciferous plants [21,25].

To explore the exon-intron structures of *BoPUB* genes, their genomic and coding sequences were submitted to GSDS 2.0 to analyze exon-intron architecture. The number of exons in each *BoPUB* gene varied from one to nineteen, and the composition of exon-intron was relatively conserved in the same subfamily (Figure 2c). Fourteen genes (88%) in Group A only contained one exon; almost all the members (95%) in Group B-D carried one to six exons; twelve members (60%) in Group E included seven to nine exons; five members (71%) in Group F had two exons (Figure 2c).

### 2.3. Chromosomal Localization and Duplication of BoPUB Genes

To understand the chromosomal localization of *BoPUB* genes, the information on their genome positions was extracted from the cabbage genome database and shown in the genome map (Figure 3 and Appendix A). The *BoPUB* gene family members were distributed among all nine chromosomes (C01–09). There were nine *BoPUB* genes on C01, three on C02, fourteen on C03, nine on C04, nine on C05, three on C06, six on C07, six on C08, and six on C09 (Figure 3 and Appendix A). The formation of gene family is determined by gene duplication events, including tandem duplication, segmental duplication, and whole-genome duplication [34]. To reveal the duplication process of *BoPUB* genes, the synteny analysis of these genes was carried out by the BLAST 2.11.1+ and MCScanX software. The results displayed that segmental and tandem duplication events in the whole genome occurred in ten and three *BoPUB* gene pairs, respectively (Figure 3 and Appendix A), indicating that segmental and tandem duplication events appear to play a positive role in the expansion of the *BoPUB* gene family. A similar phenomenon is also found in the duplication of PUB genes in Chinese cabbage [21]. To evaluate the selection pressure of *BoPUB* genes, the non-synonymous substitution rate (Ka), synonymous substitution rate (Ks), and Ka/Ks ratio of each duplicate gene pair were measured. As shown in Appendix A, the Ka/Ks ratios of all 13 gene pairs were less than one, suggesting that the *BoPUB* gene family may undergo purifying selection during evolution. This is consistent with the results of previous studies on the evolution of PUB genes in *Brassica oleracea* [25].

### 2.4. Interaction Relationships between BoPUB Proteins

U-box E3 ubiquitin ligases commonly function in the form of homo/heterodimers, which can modulate their protein stability *in vivo* [13,35]. To evaluate interaction relationships between BoPUB proteins, the amino acid sequences of 65 BoPUB proteins were submitted to the STRING server. The prediction of protein interaction indicated that potential protein interactions might occur between BoPUB7 and BoPUB1/6/9/11/12/16/19/23/26/30/34/36/42/43/50/51/52/54/58/60, BoPUB19 and BoPUB32, BoPUB26 and BoPUB23/36/42/45/58/64, BoPUB36 and BoPUB23/42/58/64, and BoPUB42 and BoPUB23/45/58/64 (Figure 4 and Appendix A). These results collectively suggest that BoPUB proteins are likely to form heterodimers and coordinately function in various cellular processes as described by Hu et al. [25].

### 2.5. Cis-Regulatory Elements in the Promoters of BoPUB Genes

Cis-regulatory elements in the upstream of start codons that act as the binding sites of transcription factors are crucial for the transcriptional regulation of protein-coding genes [36]. To analyze cis-regulatory elements in the promoters of *BoPUB* genes, the 2.0 kb sequences upstream of their start codons were extracted and submitted to the PlantCARE database. In total, 43 types of cis-regulatory elements were identified in the promoters of *BoPUB* genes (Figure 5a and Appendix A). Based on the related biological processes, these cis-regulatory elements were divided into four groups: growth and development process (growth and development), including nine types of elements; phytohormone response (phytohormone) including nine types of elements; light response (light), including twenty types of elements; and stress response (stress), including five types of elements (Figure 5a and Appendix A). Notably, of the total 2310 elements, 954 (41%) and 932 (40%) belonged to light and phytohormone response groups, respectively (Figure 5b). In the light-responsive elements, G-box is the most frequent member, accounting for 33% (Figure 5a). In the phytohormone response group, a larger proportion of elements are ABRE (30%) involved in the abscisic acid (ABA) response and TGACG-motif (25%) and CGTCA-motif (24%) involved in the jasmonic acid (JA) response (Figure 5a). These data suggest that *BoPUB* genes are likely to participate in response to light, ABA, and JA. It is worthy to note that the presence of ABRE in the promoter regions of *BoPUB* genes is also reported in previous studies [25], which further confirms our speculation on the involvement of *BoPUB* genes in the ABA response.

### 2.6. Spatio-Temporal Expression Patterns of BoPUB Genes in Different Tissues

Understanding the expression of a gene in different tissues is required for unravelling its biological function in growth and development [37]. To confirm the tissue-specific expression of *BoPUB* genes, the normalized expression data (fragments per kilobase per million mapped reads, FPKM) of *BoPUB* genes in seven different tissues (callus, root, stem, leaf, bud, flower, and silique) were obtained from the NCBI Gene Expression Omnibus (GEO) datasets and shown in a log2-scaled heatmap (Figure 6). A total of 65 *BoPUB* genes were clustered into six groups (Groups I–VI) based on their transcript levels (Figure 6). As shown in Figure 6, the global expression levels of genes in Groups I and II were lower than those of the members in the other four groups. Most members in Groups III and IV were strongly expressed in calluses, roots, and stems (Figure 6). All the genes in Group V exhibited constitutive expression in all tissues (Figure 6). In Group VI, *BoPUB17* showed flower-specific expression, and *BoPUB22* and *BoPUB44* presented stem-specific expression (Figure 6). Taken together, these results indicate that *BoPUB* genes may be closely implicated in the growth and development of various tissues and organs.

### 2.7. Expression Patterns of BoPUB Genes Involved in Cuticular Wax Biosynthesis

E3 ubiquitin ligases have been widely reported to regulate cuticular wax biosynthesis in many species, including *Arabidopsis thaliana* [38,39] and *Oryza sativa* [40]. To understand whether *BoPUB* genes are involved in cuticular wax biosynthesis, the transcript levels of *BoPUB* genes in cabbage leaves from cuticular wax-deficient mutant *non-wax glossy* (*nwgl*) and its wild-type (WT) were analyzed based on RNA-seq data previously reported (Figure 7a) [41]. The results indicated that eight *BoPUB* genes were differentially expressed between *nwgl* and WT leaves (Figure 7b). Of these genes, the transcript levels of six *BoPUB* genes (*BoPUB19*, *29*, *34*, *40*, *48*, and *54*) were up-regulated, and those of two *BoPUB* genes (*BoPUB32* and *59*) were down-regulated (Figure 7b). The transcription of *BoPUB34* was strongly induced, whereas the expression of *BoPUB59* was dramatically inhibited (Figure 7b). To further validate these differentially expressed *BoPUB* genes, their expression in *nwgl* and WT leaves was analyzed using qRT-PCR. As shown in Figure 7c, the transcript levels of *BoPUB19*, *34*, and *54* were significantly elevated in *nwgl* leaves relative to its wild type (WT), consistent with previous RNA-seq analysis, implying that *BoPUB* genes may participate in cuticular wax biosynthesis.

### 2.8. Expression Patterns of BoPUB Genes in Response to Cold Stress

PUBs have been reported to be involved in tolerance to cold stress in many species, including *Arabidopsis thaliana* [42], *Oryza sativa* [43], *Medicago truncatula* [44], and *Vitis vinifera* [45]. To better understand the roles of *BoPUB* genes in response to cold stress, the relative expression of *BoPUB* genes in cabbage leaves under low-temperature conditions was analyzed using qRT-PCR. The results showed that the expression levels of 21 *BoPUB* genes significantly altered in response to cold treatment (Figure 8). Of these genes, the transcript levels of 15 *BoPUB* genes (*BoPUB3*, *6*, *7*, *8*, *9*, *10*, *11*, *15*, *31*, *50*, *56*, *58*, *62*, *63*, and *65*) exhibited an increased expression trend over 24 h after low-temperature treatment, and the expression of six *BoPUB* genes (*BoPUB28*, *45*, *46*, *51*, *60*, and *61*) was significantly inhibited (Figure 8). Furthermore, we also found that the transcript levels of some genes (*BoPUB6*, *10*, *15*, *31*, and *65*) peaked at 6 h or 12 h after cold treatment and subsequently decreased at 24 h after cold treatment, whereas the expression of some other genes (*BoPUB8*, *11*, and *62*) was gradually up-regulated over 24 h after cold treatment and peaked at 24 h after cold treatment. Notably, four *BoPUB* genes (*BoPUB8*, *11*, *50*, and *56*) showed dramatically up-regulated expression under low-temperature conditions. These results reveal that *BoPUB* genes may be involved in response to cold stress.

### 2.9. Expression Patterns of BoPUB Genes in Response to Alternaria brassicicola Infection

A large amount of evidence has demonstrated that PUBs can serve as important regulators in resistance to pathogens [46,47,48]. To assess whether *BoPUB* genes take part in response to pathogen infection, we analyzed the expression patterns of *BoPUB* genes in cabbage leaves after *Alternaria brassicicola* infection based on the microarray gene expression data previously reported [49]. The results showed that 65 *BoPUB* genes were clustered into five groups (Groups a–e) according to their expression levels (Figure 9). In Group a, the transcription of *BoPUB8 and BoPUB54* was induced in leaves at 12 and 24 h post-infection (hpi), whereas their expression decreased at 48 hpi. After *Alternaria brassicicola* treatment, the differential expressions of genes in Groups b and c were relatively weak and showed no obvious regularity (Figure 9). The transcript levels of many genes in Group d exhibited an increased trend in leaves at 12 and 24 hpi, including *BoPUB5*, *14*, *20*, *22*, *23*, *26*, *30*, *31*, *42*, *43*, *44*, *58*, and *63* (Figure 9). Almost all members in Group e were strongly expressed in leaves at 48 hpi, especially *BoPUB16*, *19*, *29*, *51*, and *54* (Figure 9). These results imply that *BoPUB* genes may participate in response to pathogen infection.

## 3. Discussion

Although PUBs are structurally conserved in many species, their gene family size difference still exists across different species. To date, 56 PUB genes in *Vitis vinifera* [22], 62 in *Solanum lycopersicum* [23], 62 in *Pyrus bretschneideri* [24], 64 in *Medicago truncatula* [44], 64 in *Arabidopsis thaliana* [50], 77 in *Oryza sativa* [51], 101 in *Brassica rapa* [21], 125 in *Glycine max* [52], 93–208 in *Gossypium* species [53], and 213 in *Triticum aestivum* [54] have been found. In the present study, 65 PUB genes were confidently found in the cabbage genome. However, only two and twenty-one U-box E3 ubiquitin ligase genes are identified in *Saccharomyces cerevisiae* and *Homo sapiens*, respectively [7]. This evidence suggests that the U-box E3 ubiquitin ligase gene family size in plants appears to be larger than that in fungi and mammals, which is probably because plant U-box E3 ubiquitin ligase genes undergo more gene duplication events, especially whole-genome duplication that results in the formation of polyploidy. Gene duplication is a major force for the expansion of gene family members and mainly includes three types: segmental duplication, tandem duplication, and whole-genome duplication [34]. Synteny analysis revealed that thirteen gene duplication events, including ten segmental duplication events and three tandem duplication events, occurred in the *BoPUB* gene family, indicating that segmental and tandem duplication events contribute to the amplification of the *BoPUB* gene family. On the other hand, gene duplication also provides a foundation for the functional diversity of U-box E3 ubiquitin ligases in cabbage. Similar to *Vitis vinifera* [22] and *Solanum lycopersicum* [23], there are also a large number of intronless genes in the *BoPUB* gene family, and these PUB genes generally only contain a U-box domain, such as the members in Group A (Figure 2a,b). The insertion of introns during evolution increases gene length and introduces additional motifs, thereby leading to the sub-functionalization of PUBs to undertake more sophisticated and versatile functions in various signaling pathways. Based on the variety of additional motifs, the PUB protein family is generally divided into five clusters, including U-box, U-box-Arm, U-box-Kinase, Ufd2P_core-U-box, and U-box-WD40/Kinesin-associated protein (KAP)/Toll/interleukin-1 receptor (TIR) [23]. Most motifs related to protein interaction generally facilitate the regulation of substrate recognition and ubiquitination, homo/heterodimerization, protein stability, and enzyme activity. In addition to these common motifs, we have also identified several unique motifs (Arf, Usp, and CLTH) in the BoPUB protein family, which may confer new functions to their proteins in some specific regulatory pathways.

PUBs are a vital class of E3 ubiquitin ligases in plants and are reported to be involved in many biological processes [13]. The analysis of cis-regulatory elements indicated that *BoPUB* genes may be widely involved in growth and development, phytohormone, light, and stress responses, especially light, ABA, and JA signaling pathways. Despite the fact that the function of PUBs in the light response remains elusive, the relationships between PUBs and ABA biosynthesis and signal transduction have been well characterized. AtPUB44/SAUL1 mediates the ubiquitination and degradation of abscisic aldehyde oxidase 3 (AAO3) to affect ABA biosynthesis [55]. Furthermore, CHIP, a PUB in *Arabidopsis*, is likely to be involved in the modulation of the ABA response by targeting phosphatase 2A (PP2A), an essential component in the ABA signaling pathway [56]. Loss of AtPUB12/13 function results in ABA-insensitive phenotypes via the ubiquitination of ABA co-receptor ABA-INSENSITIVE 1 (ABI1) [57], whereas AtPUB9, 18, and 19 can act as negative regulators in response to ABA [58,59]. In this study, the detection of ABA-responsive elements in the promotors of *BoPUB* genes suggests that the expression of PUB genes appears to be induced by ABA in cabbage. Thus, the notion of whether BoPUB proteins function in the feedback regulation of the ABA signaling pathway still deserves to be further explored. In recent years, a lot of documents with regard to the role of PUB proteins in the JA signaling pathway have gradually emerged. AtPUB10 reduces the stability of MYC2, a key transcription factor in the JA signaling pathway, through the ubiquitin-proteasome system [60]. Moreover, the expression of *Arabidopsis* U-box senescence related 1 (AtUSR1), another PUB protein, is activated by the MYC2-mediated JA signaling pathway in the process of leaf senescence [61]. These findings imply that BoPUB proteins might also be implicated in the JA response by targeting MYC transcription factors.

As a typical abiotic stress, drought can severely damage plant growth, development, and crop yield [62]. Most studies have demonstrated that cuticular wax, an indispensable hydrophobic layer that covers the aboveground surface of plants, can play a positive role in tolerance to drought stress. Significant progress has been made in the last decade uncovering the function of E3 ubiquitin ligases in the regulation of cuticular wax biosynthesis in response to drought stress. For example, a RING-type E3 ligase CER9 negatively regulates cuticular wax biosynthesis to affect the drought tolerance of *Arabidopsis* plants [63]. Overexpression of DHS, a RING-type E3 ubiquitin ligase, reduces tolerance to drought stress by controlling cuticular wax biosynthesis in rice plants [40]. Moreover, suppression of *SAGL1*, a gene encoding the F-box subunit of SCF E3 ligase complex, increases cuticular wax accumulation and drought tolerance in *Arabidopsis* [64]. So far, few PUBs have been reported to be involved in cuticular wax biosynthesis. In this study, we detected eight differentially expressed *BoPUB* genes in a cuticular wax-deficient mutant based on RNA-seq data, suggesting that *BoPUB* genes may play an important role in cuticular wax biosynthesis. Meanwhile, we also confirm that the expression levels of three *BoPUB* genes were significantly up-regulated in a cuticular wax-deficient mutant compared to its wild type, which reflects that BoPUB proteins might act as repressors in the biosynthesis of cuticular wax consistent with other types of E3 ligases previously reported.

Low temperature is one of the major abiotic stresses for plant growth and development [65]. In the last decade, the roles of PUBs in response to cold stress have been well characterized in many different species. Most studies have revealed that PUBs can serve as positive regulators in cold tolerance. For example, AtPUB25/26 enhances freezing tolerance in *Arabidopsis thaliana* by accelerating the ubiquitination degradation of MYB15, a transcriptional repressor in the cold signaling pathway [42]. Overexpression of OsPUB2 and OsPUB3 also confers better tolerance to cold stress in rice plants [43]. Similarly, PUBs from horticultural plants, such as *Vitis amurensis* and *Capsicum annuum*, are also reported to enhance plant cold stress tolerance [45,66]. Nevertheless, AtCHIP-overexpressing *Arabidopsis* plants behave more sensitive to low-temperature treatment [67]. In this work, we identified 21 differentially expressed *BoPUB* genes in cabbage leaves after low-temperature treatment, suggesting that *BoPUB* genes are likely to play an important role in resistance to cold stress. Moreover, we also noticed that these *BoPUB* genes displayed many different expression patterns in leaves exposed to low temperatures, indicating that BoPUB proteins might play multiple roles in the regulation of cold resistance.

Plant innate immunity is a complex process, in which numerous cell-surface and intracellular receptors recognize various signals related to infection and subsequently transmit them to downstream components, thus leading to the activation of defensive responses to pathogens [68]. Based on the variety of receptors that trigger immune responses, plant innate immunity is divided into two types: pathogen-associated molecular patterns (PAMPs)-triggered immunity (PTI) that results from pattern recognition receptors (PRRs) on the plasma membrane (PM) and effector-triggered immunity (ETI) that is triggered by intracellular receptors encoded by resistance (R) genes [68]. Previous evidence has demonstrated that PUBs play a vital role in the regulation of immune responses. Inactivation of *AtPUB12, 13, 22*, *23*, and *24* can promote the PTI response triggered by PM-located receptors [46,47,48], and loss of *MUSE3*, an additional *Arabidopsis* U-box E3 ubiquitin ligase gene, enhances the R protein-mediated immunity [69], indicating that some PUBs are negatively correlated to immune responses. However, many PUBs are also reported to be the positive regulators of immune responses. For instance, overexpression of *AtPUB44/SAUL1* leads to PTI-associated autoimmunity in the wild-type background [70], while the other two highly homologous U-box proteins, MAC3A and MAC3B, are essential for the ETI-related immune response in *Arabidopsis* [71]. In the present study, we found that the expression of *BoPUB* genes was mainly up-regulated in response to *Alternaria brassicicola* infection, suggesting that PUBs in cabbage appear to play a positive role in the resistance to pathogen infection. Similarly, we also detected some up-regulated *BoPUB* genes in the microarray gene expression data of cabbage seedlings infected by *Xanthomonas campestris* pv. *campestris* (Appendix A) [72]. It is therefore conceivable that the activation of *BoPUB* genes might be involved in a broad-spectrum resistance to different pathogens.

## 4. Materials and Methods

### 4.1. Plant Materials and Cold Treatment

Seeds of cabbage (*Brassica oleracea* var. *capitata*) and its wax-deficient mutant used for this study were obtained from the *Brassica oleracea* genetics and breeding laboratory (Northeast Agricultural University, Harbin, China). Cabbage seedlings were grown in a culture chamber under standard culture conditions. For cold treatment, cabbage seedlings at the four-leaf stage were incubated at 4 °C for 0 h, 6 h, 12 h, and 24 h, respectively. Leaves were collected from cabbage seedlings at different time points after cold treatment and immediately frozen and stored at 80 °C until use.

### 4.2. Identification of BoPUB Genes

To identify U-box E3 ubiquitin ligase genes in cabbage, the amino acid sequence of AtCHIP (AT3G07370) was downloaded from The Arabidopsis Information Resource (TAIR, https://www.arabidopsis.org/index.jsp accessed on 10 March 2022) [73] and retrieved against the *Brassica oleracea* var. *capitata*_446_v1.0 protein database in Phytozome v12.1 (https://phytozome.jgi.doe.gov/pz/portal.html accessed on 10 March 2022) [74] using the BLASTP program with default parameters [75]. The HMM profiles of U-box domain (PF04564.15) were obtained from the Pfam database (http://pfam.xfam.org/ accessed on 20 March 2022) [76] and used to search U-box domain-contained proteins in *Brassica oleracea* var. *capitata* using the HMMER 3.0 software with default parameters [77]. All the redundant proteins were removed, and the remaining proteins were submitted to the SMART (http://smart.embl-heidelberg.de/ accessed on 10 March 2022) [78] and CDD databases (https://www.ncbi.nlm.nih.gov/Structure/cdd/ accessed on 10 March 2022) [79] to further confirm the presence of U-box domain. The number of amino acids, molecular weight (MW), and isoelectric point (pI) of BoPUB proteins were analyzed using ProtParam and ExPasy-Compute pI/Mw [80]. The subcellular localization of BoPUB proteins was predicted by the WoLF PSORT server (https://wolfpsort.hgc.jp/ accessed on 10 March 2022) [81].

### 4.3. Phylogenetic Analysis

For the phylogenetic analysis of *BoPUB* genes, the sequence alignment of their proteins was carried out by Clustal Omega (https://www.ebi.ac.uk/Tools/msa/clustalo/ accessed on 10 March 2022) [82], and then the alignment results were used to construct a phylogenetic tree with the neighbor-joining (NJ) method and 1000 bootstrap replicates by the MEGA 11.0 software [83].

### 4.4. Conserved Domain and Gene Structure Analysis

The conserved domains of BoPUB proteins were analyzed by the Pfam database (http://pfam.xfam.org/ accessed on 20 March 2022) [76] and visualized by the TBtools v1.095 software [84]. The exon-intron structures of *BoPUB* genes were visualized by the Gene Structure Display Server 2.0 (GSDS 2.0, http://gsds.cbi.-pku.edu.cn/ accessed on 10 March 2022) [85].

### 4.5. Chromosomal Localization and Synteny Analysis

The chromosomal localization information of *BoPUB* genes was extracted from the *Brassica oleracea* var. *capitata*_446_v1.0 genome database and visualized by the TBtools software [84]. The synteny analysis of *BoPUB* genes was performed as previously described by Song et al. [86] with some modifications. Briefly, BoPUB protein sequences were analyzed by the BLASTP [75] and MCScanX [87] programs, and the syntenic relationships of *BoPUB* genes were shown by the TBtools software [84]. For the analysis of selection pressure on duplicate *BoPUB* gene pairs, the coding sequences of *BoPUB* genes were used to calculate the Ka, Ks, and Ka/Ks ratio of each duplicate gene pair using the KaKs_Calculator 2.0 software [88].

### 4.6. Protein Interaction Prediction

For the prediction of interactions between BoPUB proteins, the amino acid sequences of BoPUB proteins were downloaded from the *Brassica oleracea* var. *capitata*_446_v1.0 protein database and submitted to the STRING server (https://string-db.org/ accessed on 20 March 2022) [89].

### 4.7. Cis-Regulatory Element Prediction

For the prediction of cis-regulatory elements in the promoters of *BoPUB* genes, the promoter sequences (2.0 kb sequence upstream of start codons) of *BoPUB* genes were extracted from the *Brassica oleracea* var. *capitata*_446_v1.0 genome database and submitted to the PlantCARE database (http://bioinformatics.psb.ugent.be/webtools/plantcare/html/ accessed on 20 March 2022) [90].

### 4.8. Expression Analysis of BoPUB Genes in Different Tissues

The expression data (FPKM) for *BoPUB* genes in seven tissues (callus, root, stem, leaf, bud, flower, and silique) from *Brassica oleracea* var. *capitata* were extracted from the NCBI GEO DataSets (accession number: GSE42891, https://www.ncbi.nlm.nih.gov/geo/query/acc.cgi?acc=GSE42891 accessed on 20 May 2022) [26,91]. The log2-transformed expression profiles of *BoPUB* genes were hierarchically clustered and visualized by the heatmap package in the TBtools v1.095 software [84].

### 4.9. Expression Analysis of BoPUB Genes Involved in Cuticular Wax Biosynthesis

The expression data (FPKM) of *BoPUB* genes in cabbage leaves from cuticular wax-deficient mutant *nwgl* and its wild type (WT) were downloaded from the NCBI GEO datasets (accession number: GSE130405, https://www.ncbi.nlm.nih.gov/geo/query/acc.cgi?acc=GSE130405 accessed on 20 May 2022) [41]. Fold changes in the expression levels of *BoPUB* genes between *nwgl* and WT leaves were calculated. *BoPUB* genes with fold change of more than 2 (up-regulation) or less than 0.5 (down-regulation) and *p* value < 0.05 are considered to be differentially expressed.

### 4.10. RNA Isolation and Quantitative Real-Time PCR

Total RNA was extracted from cabbage leaves at the four-leaf stage using the FastPure Plant Total RNA Isolation Kit (Nanjing, China, Vazyme) according to the manufacturer’s instructions. Genome DNA digestion and reverse transcription of the extracted RNA were carried out using the HiScript III 1st Strand cDNA Synthesis Kit (+gDNA wiper) (Nanjing, China, Vazyme) according to the manufacturer’s protocol. Quantitative real-time PCR (qRT-PCR) was carried out with the HiScript II One Step qRT-PCR SYBR Green Kit (Nanjing, China, Vazyme) using the StepOne Plus Real-Time PCR System (Applied Biosystems). PCR primers listed in Appendix A were designed by Primer Express 3.0. PCR reaction was performed in a volume of 20 μL with the following program: 95 °C for 3 min, followed by 40 cycles of 95 °C for 10 s and 60 °C for 30 s. The 2^−∆∆Ct^ method was used to calculate relative gene expression levels [92]. *BoTUB6* was used as a reference gene [93]. Three independent biological replicates with three technical repeats each were conducted.

### 4.11. Expression Analysis of BoPUB Genes after Alternaria brassicicola Infection

The *Arabidopsis* oligonucleotide microarray-based gene expression data of cabbage leaves at 0, 12, 24, and 48 h post-infection (hpi) by *Alternaria brassicicola* were downloaded from the NCBI GEO datasets (accession number: GSE155051, https://www.ncbi.nlm.nih.gov/geo/query/acc.cgi?acc=GSE155051 accessed on 20 May 2022) [49]. The probe set IDs corresponding to *BoPUB* genes were retrieved against the gene expression data downloaded above. The values representing fold changes in the expression levels of *BoPUB* genes between pathogen-treated samples and controls were extracted, hierarchically clustered, and visualized in a log2-scaled heatmap by the TBtools v1.095 software [84].

### 4.12. Statistical Analysis

All statistical analyses were conducted by Microsoft Excel 2019 and GraphPad Prism 8.0 software. The data were shown as the means ± standard deviation (SD) of three independent biological experiments. Two-tailed Student’s *t*-test and one-way ANOVA were used to assess the significance levels of data between samples (* *p* < 0.05, ** *p* < 0.01).

## 5. Conclusions

Using in silico analysis, we identified 65 PUB genes in the cabbage genome. The *BoPUB* gene family members were divided into six clusters based on their evolutionary relationships. Segmental and tandem duplication events were found to contribute to the expansion of the *BoPUB* gene family. We revealed that BoPUB proteins are able to form heterodimers, and their genes are closely related to the growth and development of various tissues and organs. We found that *BoPUB* genes are involved in cuticular wax biosynthesis, cold response, and immune response. Our findings provide better guidance for understanding the function of PUB genes in cabbage.

## Figures and Tables

**Figure 1 plants-12-01437-f001:**
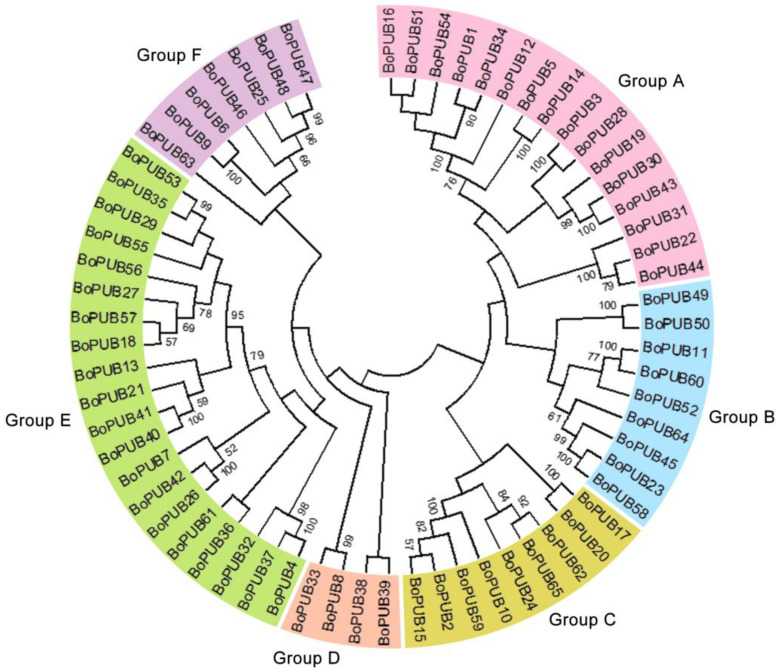
Phylogenetic relationships of *BoPUB* genes. The sequence alignment of 65 BoPUB proteins was conducted by Clustal Omega. A phylogenetic tree based on the alignment above was constructed by the neighbor-joining (NJ) method with 1000 bootstrap replicates. The *BoPUB* gene family members are divided into six groups (Group A–F). The genes in different groups are highlighted in six different colors, respectively.

**Figure 2 plants-12-01437-f002:**
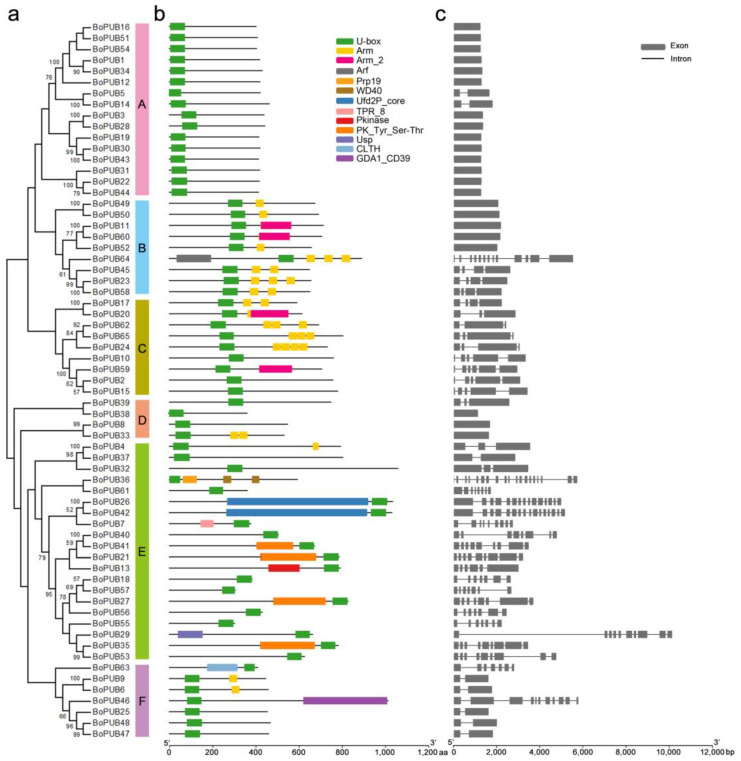
Phylogenetic relationships, conserved domain composition, and exon-intron structures of *BoPUB* genes. (**a**) Phylogenetic relationships of *BoPUB* genes. A phylogenetic tree based on the sequence alignment of BoPUB proteins was constructed by the MEGA 11.0 software, as described in Figure 1. Groups A–F are highlighted in six different colors corresponding to Figure 1, respectively. (**b**) Distribution of conserved domains in the BoPUB protein sequences. Sixty-five BoPUB protein sequences were submitted to the Pfam database to analyze the conserved domains of BoPUB proteins. The composition of conserved domains in the BoPUB protein sequences was visualized by the TBtools v1.095 software. Thirteen conserved domains marked in different colors are shown. Pfam accession number: U-box domain (PF04564.15), armadillo (Arm) repeat (PF00514.23), Arm_2 repeat (PF04826.13), ADP ribosylation factor (Arf) domain (PF00025.23), Prp19 domain (PF08606.13), WD40 repeat (PF00400.32), ubiquitin elongating factor core (Ufd2P_core) domain (PF10408.9), tetratricopeptide repeat (TPR)_8 domain (PF13181.8), protein kinase (Pkinase) domain (PF00069.25), protein tyrosine and serine/threonine kinase (PK_Tyr_Ser-Thr) domain (PF07714.19), universal stress protein (Usp) domain (PF00582.28), CTLH/CRA C-terminal to LisH motif (CTLH) domain (PF10607.11), and GDA1/CD39 nucleoside phosphatase (GDA1_CD39) domain (PF01150.19). (**c**) Exon-intron structures of *BoPUB* genes. The genomic and coding sequences of *BoPUB* genes were submitted to the Gene Structure Display Server to analyze the exon-intron structures of *BoPUB* genes. Grey bars indicate exons, and black lines indicate introns.

**Figure 3 plants-12-01437-f003:**
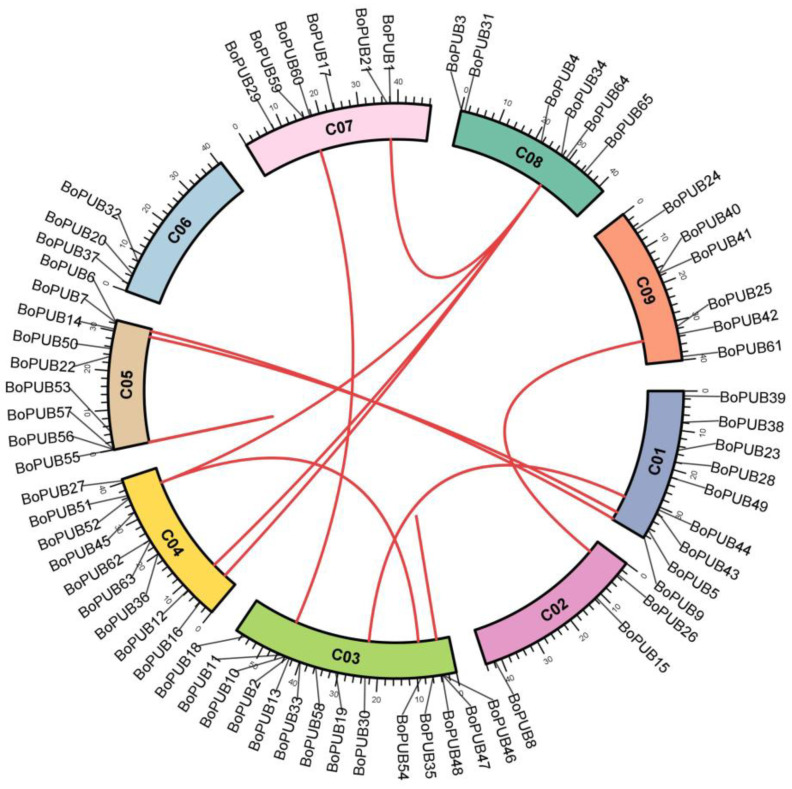
Distribution and duplication of *BoPUB* genes. Nine chromosomes (C01–09) in cabbage are arranged in a circular pattern, and the location of 65 *BoPUB* genes is marked. Red lines represent the homologous relationships of *BoPUB* genes.

**Figure 4 plants-12-01437-f004:**
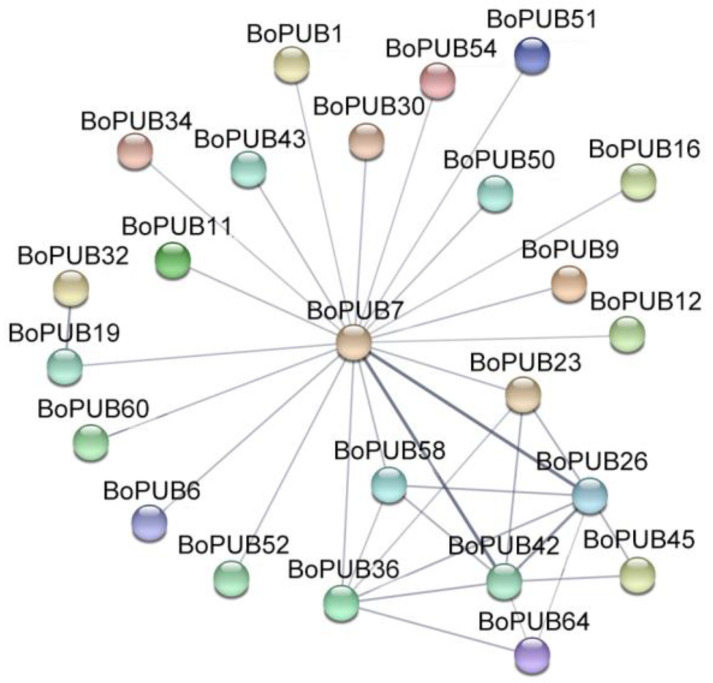
Prediction of interactions between BoPUB proteins. Grey lines indicate putative interactions between BoPUB proteins, and the depth of line color represents the possibility of protein–protein interaction. Three-dimensional spheres represent BoPUB proteins.

**Figure 5 plants-12-01437-f005:**
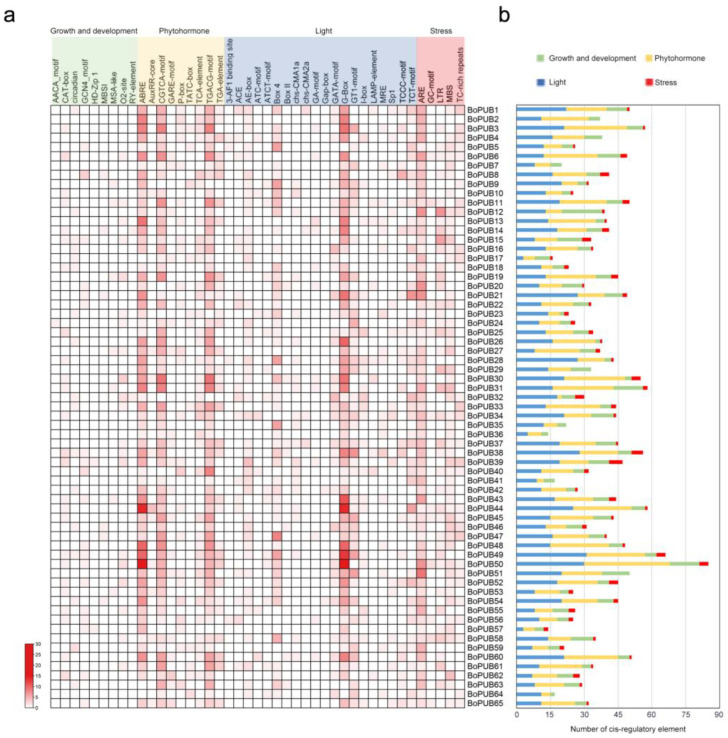
Analysis of cis-regulatory elements in the promoter regions of *BoPUB* genes. (**a**) Heatmap showing the number of individual cis-regulatory elements in the promoter of each *BoPUB* gene. Each row and column in the red grid indicate a single gene and cis-regulatory element, respectively, and their names are shown. Forty-three cis-regulatory elements are clustered into four categories based on corresponding biological processes, including growth and development process (growth and development), phytohormone response (phytohormone), light response (light), and stress response (stress). Detailed information on these cis-regulatory elements is listed in Appendix A. (**b**) Stacked histogram showing the composition of four types of cis-regulatory elements in the promoter of each *BoPUB* gene. The number of cis-regulatory elements is shown on the *x*-axis, and gene names are shown on the *y*-axis. Green bars represent the growth and development process (growth and development); yellow bars represent the phytohormone response (phytohormone); blue bars represent the light response (light); and red bars represent the stress response (stress).

**Figure 6 plants-12-01437-f006:**
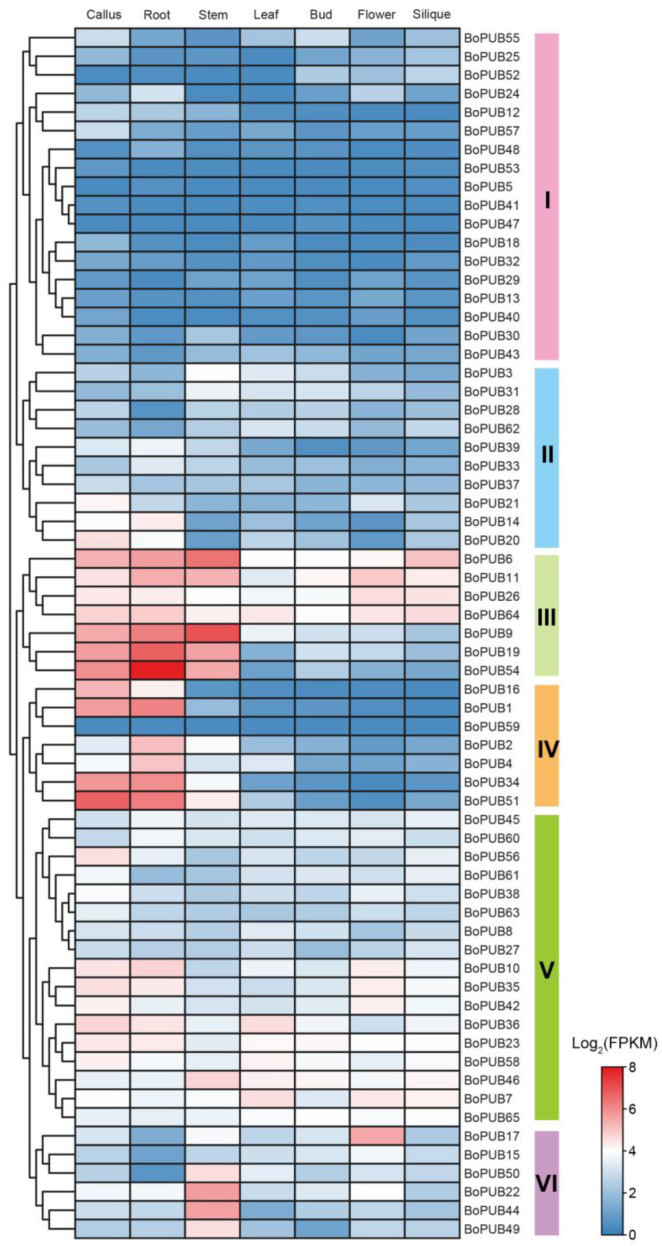
Expression profiles of *BoPUB* genes in various tissues and organs. The normalized expression data, as fragments per kilobase per million mapped reads (FPKM), of *BoPUB* genes in seven tissues, including callus, root, stem, leaf, bud, flower, and silique, were obtained from the RNA-seq data of cabbage (*Brassica oleracea* var. *capitata*). The expression levels of *BoPUB* genes were plotted in a log2-scaled heatmap. Each row and column in the color heatmap indicate a single gene and tissue, respectively, and their names are shown. Sixty-five *BoPUB* genes are hierarchically clustered into six groups (Groups I–VI) according to their expression levels. Groups I–VI are highlighted in six different colors, respectively.

**Figure 7 plants-12-01437-f007:**
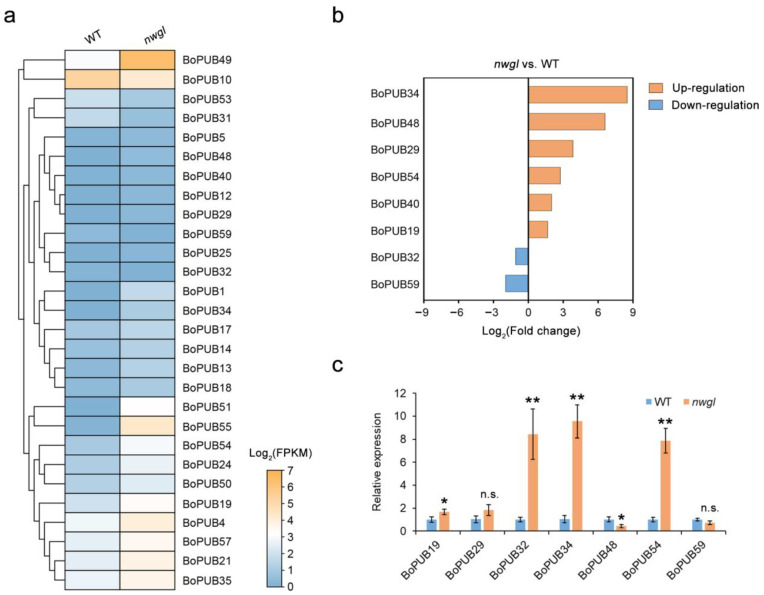
Differential expression of *BoPUB* genes between wild type and *nwgl* mutant. (**a**) Expression profiles of *BoPUB* genes in wild type and *nwgl* mutant. The normalized expression data, as fragments per kilobase per million mapped reads (FPKM), of *BoPUB* genes in wild type (WT) and *nwgl* leaves were obtained from the RNA-seq data of cabbage (*Brassica oleracea* var. *capitata*). The expression levels of *BoPUB* genes were plotted in a log2-scaled heatmap. Each row in the color heatmap indicates a single gene, and their names are shown. (**b**) Differentially expressed *BoPUB* genes in *nwgl* relative to its wild type. The expression data (FPKM) of *BoPUB* genes were obtained from the RNA-seq data of *nwgl* and its wild type (WT). Fold changes in the expression levels of *BoPUB* genes were shown in a log2-scaled histogram. Gene names are shown on the y-axis, and the log2-transformed fold changes in gene expression levels are shown on the x-axis. Orange bars represent up-regulated *BoPUB* genes, and blue bars represent down-regulated *BoPUB* genes. (**c**) The relative expression of *BoPUB* genes in wild type and *nwgl* mutant. Total RNA was extracted and submitted to quantitative real-time PCR. The *BoTUB6* gene was used as the internal control. Error bars represent the means ± standard deviation (SD) of three independent experiments. Asterisks indicate significant differences (* *p* <  0.05, ** *p* < 0.01; two-tailed Student’s *t*-test). n.s., not significant.

**Figure 8 plants-12-01437-f008:**
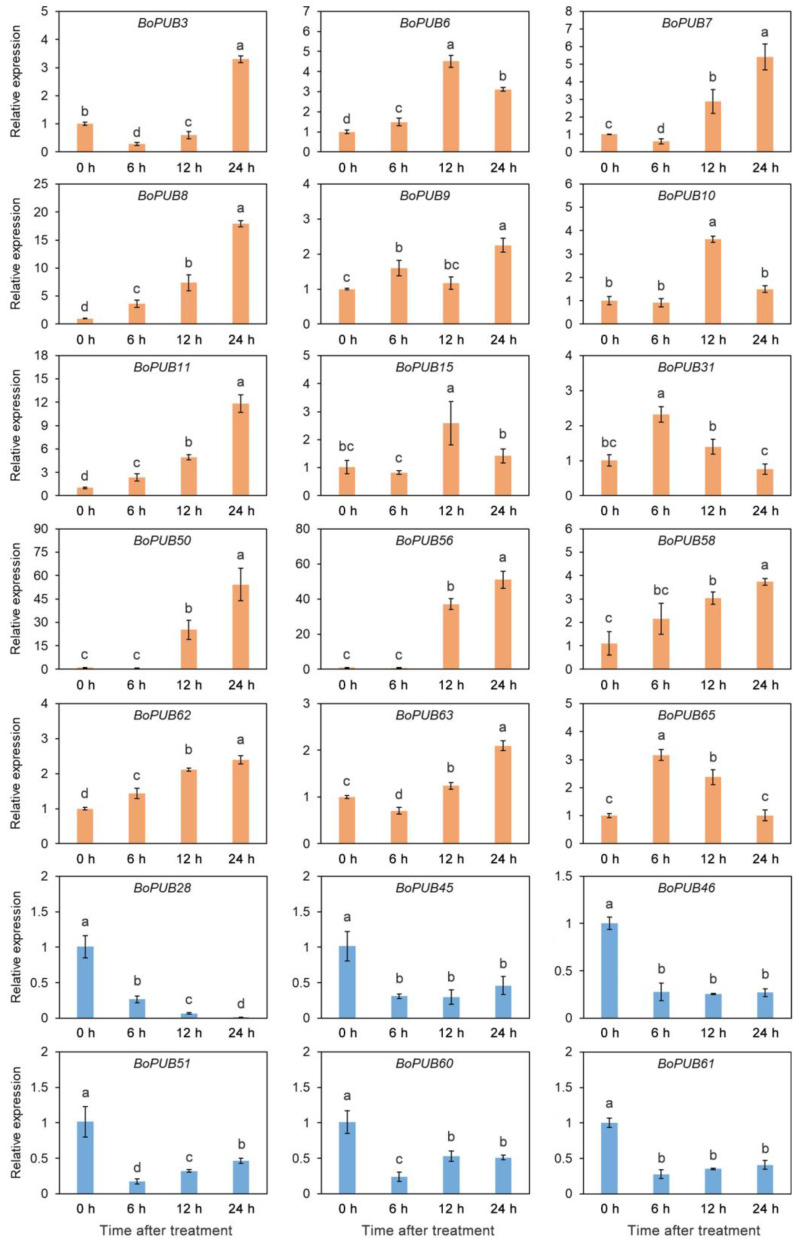
Expression profiles of *BoPUB* genes under low-temperature treatment. Total RNA was isolated from cabbage leaves after 4 ℃ treatment for 0 h, 6 h, 12 h, and 24 h, respectively. The extracted total RNA was then submitted to quantitative real-time PCR. The *BoTUB6* gene was used as the internal control. Error bars represent the means ± standard deviation (SD) of three independent experiments. Significant differences were determined by Duncan’s multiple range test (*p* < 0.05; one-way ANOVA). The lowercase letters above each bar indicate significant differences. Bars with the same lowercase letter represent no significant difference, and vice versa.

**Figure 9 plants-12-01437-f009:**
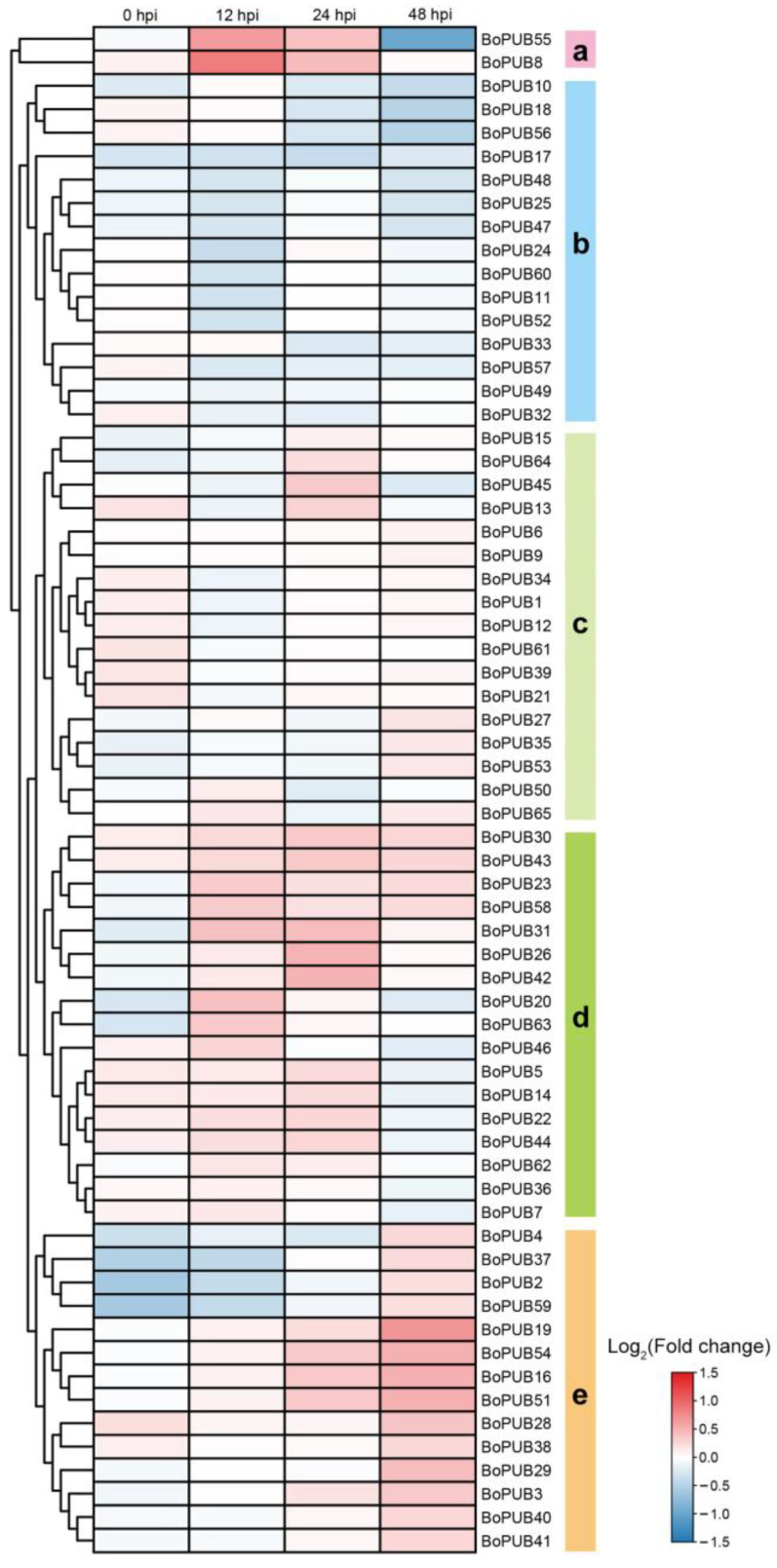
Expression profiles of *BoPUB* genes after *Alternaria brassicicola* infection. The expression data of *BoPUB* genes were obtained from the microarray gene expression data of cabbage leaves at 0, 12, 24, and 48 h post-infection (hpi) by *Alternaria brassicicola*. Fold changes in the expression levels of *BoPUB* genes were plotted in a log2-scaled heatmap. Each row and column in the color heatmap indicate a single gene and time point, respectively, and their names are shown. *BoPUB* genes are hierarchically clustered into five groups (Groups a–e) according to their expression levels. Groups a to e are highlighted in five different colors, respectively.

## Data Availability

The data supporting the findings of this study are included in this article and its Appendix A. The datasets analyzed in this study are available in the Gene Expression Omnibus (GEO) repository (accession number: GSE42891, GSE130405, GSE155051, and GSE68670).

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
