# Peer review of "Genome-Wide Identification of the U-Box E3 Ubiquitin Ligase Gene Family in Cabbage (Brassica oleracea var. capitata) and Its Expression Analysis in Response to Cold Stress and Pathogen Infection"

_plants, 2023, doi:10.3390/plants12071437_

Round 1

Reviewer 1 Report

The manuscript by Wang et. al. entitled “Genome-wide identification and expression analysis of the U-box E3 ubiquitin ligase gene family in Brassica oleracea” reports that expression of the U-box E3 ubiquitin ligase gene family of B. oleracea in response to stresses. The work is technically sound piece of research, and within the scope of Plants, but it requires major revision before its acceptance for publication.

1.      Provide more references in the Introduction, such as Hu D, Xie Q, Liu Q, Zuo T, Zhang H, Zhang Y, Lian X, Zhu L. Genome-Wide Distribution, Expression and Function Analysis of the U-Box Gene Family in Brassica oleracea L. Genes (Basel). 2019 Dec 2;10(12):1000. doi: 10.3390/genes10121000. PMID: 31810369; PMCID: PMC6947298.

2.      Explain the contents related to the published data.

3.      The title could be rewritten to match with the contents.

Reviewer 2 Report

I checked your manuscript and described comment below.

Brassica oleracea var. capitata is an important vegetable originating from Brassica oleracea var. oleracea.

In this paper, the protein with Ubox domain of Brassica oleracea var. capitata is analyzed very well.

I will describe the points I noticed below.

1.     I think the title should be changed to Brassica oleracea var. capitata instead of Brassica oleracea.

2.     If you search NCBI, I think there are 57 proteins with Ubox domain in Brassica oleracea var. oleracea. I think it would be better to make a correspondence table between BoPUB and these proteins.

3.     I think it would be better to have a table of amino acid sequences to confirm the data.

4.     MEGA7 is old software. It is now MEGA11. I think that it is better to re-analyze with MEGA11.

I don't think this paper has any major mistakes or grammatical problems.

Reviewer 3 Report

The manuscript describes the work on genome-wide identification and expression analysis of the U-box E3 ubiquitin ligase gene family in Brassica oleracea. As this gene family was not previously characterized in detail, the paper will be of significant interest to readers. In general the selected approach is appropriate and the characterization of the expression levels of various members of the family under different stresses (abiotic and biotic) serves well to confirm their importance for plant development and conferring the disease/drought/cold tolerance. The study is consciously designed and well executed, which makes for convincing discussion and conclusions.

I only have one major objection to the analysis performed and it is related to the comments on the relative importance of tandem, segmental, and whole-genome duplications. While authors correctly note that “formation of gene family is determined by gene duplication events, including tandem duplication, segmental duplication, and whole-genome duplication” (Line 175-176) they state that “segmental and tandem duplication events appear to play an essential role in the expansion of the BoPUB gene family”(Lines 180-181) and “are required for the amplification of the BoPUB gene family” (Lines 352-355). However, the present study deals only with one diploid species of the Brassica family and therefore has no power to evaluate the relative importance of these two types of duplications to whole-genome duplications. Furthermore, authors state that “evidence suggests that the U-box E3 ubiquitin ligase gene family size in plants appears to be larger than that in fungi and mammals, which is probably because plant U-box E3 ubiquitin ligase genes undergo more gene duplication events and functional differentiation to adapt to sessile organism-specific environmental stimuli” (Lines 346-349). This is highly objectionable as the mammals are only thought to consist of diploid species while in plants polyploidy (resulting from whole-genome duplications) is a very common phenomenon. Therefore, such general conclusions are not substantiated in the context of this study. To this end I propose that the text needs to be corrected to reflect the limited validity of the conclusions concerned.

There is one relatively minor issue with the use of  the verb “involve” throughout the text. Examples are : “to involve in response” (Line 59), “are reported to involve” (Line 371), “is likely to involve” (Line 378), “are reported to involve in” (Line 405), etc. These also need to be corrected for the text to appear in correct English.

Round 2

Reviewer 1 Report

The authors have been addressed my major concerns, however, may figure out the contents of previous researches.
